# When Fever Strikes Twice: A Case Report of *Streptococcus pneumoniae* Myelitis with Delayed-Onset Reactive Arthritis

**DOI:** 10.3390/idr17060147

**Published:** 2025-12-08

**Authors:** Rosario Luca Norrito, Sergio Mastrilli, Felice Fiorello, Giuseppe Taormina, Lucia Di Giorgi, Grazia Mery Anna Ruggirello, Carlo Domenico Maida, Aurelio Piazza, Fabio Cartabellotta

**Affiliations:** 1Internal Medicine Ward, Buccheri La Ferla Hospital, 90100 Palermo, Italy; fiorello.felice@fbfpa.it (F.F.); taormina.giuseppe@fbfpa.it (G.T.); cartabellotta.fabio@fbfpa.it (F.C.); 2Neurology Ward, Buccheri La Ferla Hospital, 90100 Palermo, Italy; mastrilli.sergio@fbfpa.it (S.M.); digiorgi.lucia@fbfpa.it (L.D.G.); ruggirello.graziameryanna@fbfpa.it (G.M.A.R.); piazza.aurelio@fbfpa.it (A.P.); 3Internal Medicine Ward, Sant’Elia Hospital, 93100 Caltanissetta, Italy; carlodomenico.maida@hotmail.com

**Keywords:** *S. pneumoniae*, myelitis, reactive arthritis, infectious disease

## Abstract

**Background:***Streptococcus pneumoniae* is a well-known pathogen responsible for respiratory and invasive diseases; however, central nervous system (CNS) involvement in the form of bacterial myelitis is exceedingly rare, particularly in immunocompetent adults. Moreover, the association between pneumococcal infections and reactive arthritis is scarcely documented. We report an unusual case of pneumococcal myelitis complicated by reactive arthritis in an elderly patient with no evident immunosuppression. **Case Presentation:** A 68-year-old man with a medical history of hypertension, benign prostatic hyperplasia, multiple disc herniations, and a resected pancreatic neuroendocrine tumour presented to the emergency department with acute urinary retention and fever (38.5 °C). The neurological examination revealed lower limb weakness and decreased deep tendon reflexes. Spinal magnetic resonance demonstrated T2 hyperintense lesions suggestive of longitudinally transverse myelitis. Cerebrospinal fluid (CSF) analysis showed pleocytosis with elevated protein levels; the polymerase chain reaction (PCR) test resulted positive result for *Streptococcus pneumoniae*. The patient received intravenous antimicrobial and corticosteroid therapy with partial neurological improvement. Within days, he developed acute monoarthritis of the right ankle. Joint aspiration revealed sterile inflammatory fluid, negative for crystals and cultures, supporting a diagnosis of reactive arthritis. The articular symptoms resolved with the use of prednisone. An extensive immunological work-up was negative, and no other infectious or autoimmune triggers were identified. The patient underwent a structured rehabilitation program with gradual improvement in motor function over the following weeks. **Conclusions:** This case illustrates a rare clinical scenario of pneumococcal myelitis associated with reactive arthritis in a patient without overt immunosuppression. It highlights the importance of considering bacterial aetiologies in cases of acute transverse myelitis and the potential for unusual systemic immune responses such as reactive arthritis. Early recognition and the administration of appropriate antimicrobial and supportive therapies are crucial for improving neurological and systemic outcomes. To our knowledge, this is one of the first reported cases describing the co-occurrence of these two conditions in the context of *S. pneumoniae* infection.

## 1. Background

*Streptococcus pneumoniae* is a Gram-positive encapsulated diplococcus and a leading cause of invasive bacterial infections worldwide. It is classically associated with community-acquired pneumonia, meningitis, otitis media, and bacteraemia [1]. CNS involvement most commonly presents as meningitis, while other manifestations such as brain abscess, subdural empyema, or spinal cord infections are exceedingly rare [2]. Among these, bacterial myelitis is defined as inflammation of the spinal cord resulting from direct infection. It is an infrequent entity, particularly when caused by *S. pneumoniae*. The diagnosis of bacterial myelitis is often delayed due to its non-specific clinical presentation and the rarity of the condition, which frequently mimics more common inflammatory or autoimmune aetiologies such as multiple sclerosis or idiopathic transverse myelitis [3].

In addition to its more typical clinical presentations, *Streptococcus pneumoniae* can also provoke rare but life-threatening complications. These include sepsis and disseminated intravascular coagulation, which may manifest as purpura fulminans—a rapidly progressive thrombotic and haemorrhagic disease with a high mortality rate. Although uncommon, these severe systemic manifestations highlight the broad pathogenic role of this organism, making crucial the early recognition and treatment in disseminated pneumococcal disease [4,5].

Most reported cases of bacterial myelitis occur in immunocompromised individuals or in the setting of direct extension from adjacent infected structures (e.g., vertebral osteomyelitis, epidural abscess) or after neurosurgical procedures [6]. In contrast, spontaneous hematogenous spread leading to isolated pneumococcal myelitis in an otherwise immunocompetent host is an exceptional finding, with only a few cases described in the literature to date [7,8].

Reactive arthritis (ReA) is a sterile inflammatory arthritis that typically develops 1 to 4 weeks after a bacterial infection, most commonly involving the gastrointestinal or urogenital tract [9]. It is considered part of the spondyloarthropathy spectrum and is often associated with Chlamydia trachomatis, Salmonella, Shigella, Yersinia, and Campylobacter species [10]. Although *Streptococcus pneumoniae* is not commonly implicated as a trigger of ReA, rare reports suggest that pneumococcal infections may act as a stimulus for aberrant immune responses in genetically predisposed individuals, even in the absence of joint disease [11].

The co-occurrence of bacterial myelitis and reactive arthritis in the same patient, both potentially attributable to *S. pneumoniae*, has not been previously documented in the medical literature. This case highlights a unique and diagnostically challenging clinical scenario, underscoring the importance of maintaining a broad differential diagnosis in patients presenting with acute neurological deficits and systemic symptoms, even in the absence of classical risk factors for CNS infection or reactive arthritis.

## 2. Case Presentation

A 68-year-old man with a medical history of arterial hypertension, benign prostatic hyperplasia, and multiple lumbar disc herniations presented to the emergency department with a 4-day history of high-grade fever (maximum temperature: 40 °C), urinary retention, and diffuse pain localised to the lower abdomen and both lower limbs. He denied recent trauma, surgical procedures, or infectious exposures, and was not receiving immunosuppressive therapy. The patient reported no prior history of rheumatic or autoimmune disease.

On physical examination, the patient was febrile but hemodynamically stable. Neurological assessment revealed marked motor impairment of the lower limbs, with the inability to maintain leg elevation for more than 10 s. Deep tendon reflexes were absent bilaterally in the lower limbs. Sensory examination was unremarkable, with no deficits in superficial or deep modalities. Plantar reflexes were flexor bilaterally, and there were no signs of cranial nerve involvement or meningeal irritation. Bladder catheterisation confirmed urinary retention with a residual volume greater than 500 mL.

Laboratory investigations demonstrated neutrophilic leukocytosis (WBC 22,550/mm^3^, 80% neutrophils), elevated C-reactive protein (CRP 12.8 mg/dL; normal < 0.5 mg/dL), and markedly increased procalcitonin (26 ng/mL; normal < 0.05 ng/mL), consistent with a severe systemic bacterial infection. Liver and renal function tests were within normal limits (Table 1). Based on the combination of neurological findings and systemic inflammatory markers, infectious myelitis was suspected. Lumbar puncture revealed cerebrospinal fluid (CSF) with 144 white blood cells/mm^3^ (predominantly polymorphonuclear leukocytes), elevated protein (915 mg/dL), and normal glucose (60 mg/dL; serum glucose 102 mg/dL). Gram staining was negative, but multiplex polymerase chain reaction (PCR) of the CSF tested positive for *Streptococcus pneumoniae*. Blood cultures remained negative (Table 1). High-dose intravenous ceftriaxone (2 g every 12 h) was initiated immediately, resulting in clinical and laboratory improvement. Due to technical limitations, spinal magnetic resonance imaging (MRI) with gadolinium was performed four days after initiation of antibiotic therapy. The scan demonstrated non-specific T2 hyperintensity in the thoracic spinal cord without contrast enhancement or evidence of abscess or vertebral involvement and was considered non-diagnostic (Figure 1).

After 14 days of hospitalisation, the patient was discharged with partial recovery of motor function and referred to a neurorehabilitation program.

Two weeks later, he developed fever (38.5 °C), right ankle pain, swelling, erythema, and restricted joint mobility. Laboratory tests revealed a moderately elevated CRP level (6.2 mg/dL) with a normal leukocyte count. Ultrasound examination demonstrated intra- and periarticular effusion. Arthrocentesis yielded turbid synovial fluid with more than 20,000 cells/mm^3^, negative for crystals, Gram stain, and cultures. Blood cultures, and autoimmune tests were also negative. X-ray imaging showed no erosive or structural changes in the affected joint.

A diagnosis of reactive arthritis was made based on the clinical presentation and the exclusion of septic arthritis. Treatment with oral prednisone (0.5 mg/kg/day for 21 days) resulted in complete resolution of symptoms. At follow-up, the patient remained afebrile, with no recurrence of joint or neurological symptoms, and continued rehabilitation with progressive improvement in mobility.

## 3. Discussion and Conclusions

Myelitis refers to inflammation of the spinal cord and encompasses a broad range of infectious, autoimmune, paraneoplastic, and idiopathic aetiologies. Clinically, it typically presents with varying degrees of motor, sensory, and autonomic dysfunction below the level of spinal cord involvement. Hallmark symptoms include limb weakness, sensory level disturbances, sphincter dysfunction (urinary retention or incontinence), and gait abnormalities. The pattern of neurological deficits depends on the anatomical location and extent of the lesion along the spinal cord [12].

The aetiology of myelitis can be classified into infectious (viral, bacterial, fungal, parasitic), post-infectious/parainfectious (such as acute disseminated encephalomyelitis), autoimmune (e.g., neuromyelitis optica, multiple sclerosis), and idiopathic causes. Viral pathogens (e.g., Herpes Simplex Virus, Varicella Zoster Virus, enteroviruses) are the most common infectious causes of acute transverse myelitis. In contrast, bacterial involvement is relatively rare and often secondary to hematogenous dissemination or local extension from adjacent structures [13].

Bacteria that can provoke acute transverse myelitis, although uncommon, are various and should always be suspected. Reported bacterial aetiologies include encapsulated ones such as *Streptococcus pneumoniae* and *Neisseria meningitidis*; *Haemophilus* species; Gram-positive cocci, including *Staphylococcus aureus*, β-haemolytic streptococci, and *Streptococcus agalactiae*; intracellular bacteria such as *Listeria monocytogenes*; spirochetal organisms, including *Borrelia burgdorferi* (Lyme neuroborreliosis); and zoonotic pathogens such as *Brucella* spp. *Mycobacterium tuberculosis* is also able to induce acute myelitis. The mechanism of spinal involvement can be provoked by hematogenous seeding, direct contiguous extension from vertebral osteomyelitis or epidural abscess, or immune-mediated injury following systemic infection.

Among these, Lyme neuroborreliosis deserves particular attention. Although rare, *Borrelia burgdorferi* may cause myelopathy, often in association with radiculitis or meningitis, and Lyme disease is also characterised by oligoarticular or monoarticular arthritis. Importantly, Lyme arthritis’s diagnosis is often supported by specific serologic testing or molecular detection in synovial fluid. On the other hand, reactive arthritis is a sterile post-infectious synovitis with no pathogen detection in the joint. In our patient, CSF PCR was positive for *S. pneumoniae*, Lyme testing was negative, and synovial fluid cultures were sterile—findings that favour a post-infectious reactive process rather than Lyme arthritis [14].

The pathogenesis of bacterial myelitis typically involves invasion of the spinal cord parenchyma via bloodstream seeding or contiguous spread, leading to direct tissue damage, inflammation, and oedema. Inflammatory cytokines and bacterial toxins exacerbate tissue injury and can result in rapid neurological deterioration if not promptly treated [15].

*Streptococcus pneumoniae*, a leading cause of community-acquired meningitis and bacteraemia, is an extremely rare agent of isolated spinal cord infection. Only a handful of cases of pneumococcal myelitis have been described in the literature, often in immunocompromised hosts or patients with a history of meningeal or sinus infections [16]. In our patient, no obvious local or systemic source of pneumococcal infection was identified, suggesting hematogenous dissemination with preferential tropism to spinal cord structures.

Our patient exhibited a classic subacute presentation of myelitis: urinary retention, fever, and bilateral lower limb weakness with areflexia, yet preserved sensation and absent pyramidal signs. The absence of the sensory level and Babinski sign complicated the early localisation. However, the urinary retention and bilateral lower limb involvement were suggestive of involvement of the conus medullaris or lower thoracic spinal cord.

Laboratory investigations revealed a marked systemic inflammatory response with elevated CRP (12.8 mg/dL; reference < 0.5 mg/dL) and very high procalcitonin levels (26 ng/mL; reference < 0.05 ng/mL), strongly suggestive of bacterial infection [17]. CSF analysis showed neutrophilic pleocytosis, increased protein content, and normal glucose—a profile consistent with bacterial meningo-myelitis. CSF culture was negative, likely due to early antibiotic administration; however, PCR testing confirmed *S. pneumoniae* infection, demonstrating the added diagnostic value of molecular testing in partially treated CNS infections [18].

Spinal MRI was performed four days after antibiotic initiation and did not reveal active enhancement or oedema, consistent with the known reduction in imaging sensitivity following treatment [19]. Despite this, the clinical picture and laboratory findings were sufficient for diagnosis.

The role of corticosteroids in acute myelitis is controversial and context-dependent. While high-dose intravenous corticosteroids are commonly employed in idiopathic and immune-mediated myelitides to decrease oedema, their use in the context of confirmed bacterial infection requires careful multidisciplinary consideration because of their potential immunosuppressive role. In selected cases, adjunctive steroids may be considered to reduce the inflammatory harm, but their use should be used carefully.

After antibiotic treatment and partial neurological recovery, the patient developed acute monoarthritis of the right ankle with localised oedema and erythema, but without fever or systemic signs of infection. Synovial fluid analysis demonstrated sterile inflammation, with negative bacterial culture. In the context of a recent pneumococcal infection and no alternative cause, this was consistent with ReA; although a direct causal link between *S. pneumoniae* infection and the subsequent arthritis cannot be definitively established, the temporal sequence, exclusion of other causes, sterile synovial fluid, and the clinical response to corticosteroids support a post-infectious immune mechanism.

ReA is a sterile inflammatory arthritis triggered by a distant infection, classically involving the urogenital or gastrointestinal tract. However, respiratory infections—including those caused by *Streptococcus pneumoniae*—have also been implicated, though infrequently [20]. The underlying immunopathology of ReA involves cross-reactivity between microbial antigens and host tissues, molecular mimicry, and the deposition of immune complexes. HLA-B27 positivity is a well-known risk factor, although our patient tested negative, reinforcing that ReA can occur outside of the spondyloarthropathy spectrum [21].

Corticosteroid therapy (prednisone 0.5 mg/kg/day) led to complete clinical remission of the arthritis, supporting its immune-mediated origin. At the six-month follow-up, the patient showed no motor deficiency with only mild residual urinary urgency, suggesting a favourable long-term prognosis.

This case provides multiple key insights:Myelitis should be considered in patients with febrile paraparesis or urinary retention, especially when imaging or clinical examination is inconclusive.Pneumococcal myelitis is an infrequent entity, but it should be included in the differential diagnosis of acute spinal cord dysfunction.PCR testing of CSF is indispensable for identifying bacterial pathogens in patients with negative cultures, especially those who have been pretreated with antibiotics.MRI findings can be non-specific, particularly when performed after the initiation of therapy, highlighting the importance of early imaging and a strong clinical suspicion.Reactive arthritis can follow CNS bacterial infections, including pneumococcal myelitis, and should be considered in cases of sterile monoarthritis occurring after convalescence.Prompt diagnosis and multidisciplinary management involving neurologists, infectious disease specialists, and rehabilitation professionals are essential for improving both functional and systemic outcomes.


## Figures and Tables

**Figure 1 idr-17-00147-f001:**
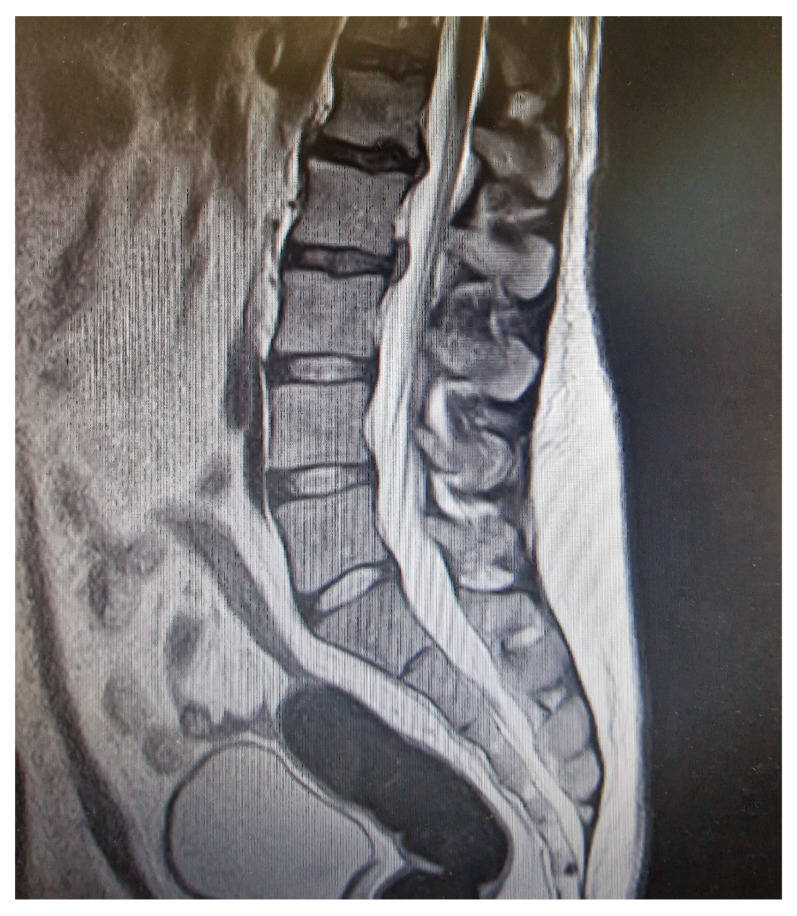
Spinal MRI was performed after 4 days of treatment and showed non-significant findings.

**Table 1 idr-17-00147-t001:** Laboratory findings at admission.

White blood cells	22,550 mm^3^
Neutrophils	80%
Platelets	233,000 μL
Haemoglobin	11.4 g/dL
C-reactive protein	12.8 mg/dL; normal < 0.5 mg/dL
Procalcitonin	26 ng/mL; normal < 0.05 ng/mL
Creatinine/clearance	0.95 mg/dL/72 mL/min
Sodium/Potassium	133/4.28 mEq/L
GOT/AST	19 U/L
GPT/ALT	30 U/L
Total bilirubin	1.06 mg/dL
INR	1.03
WBC in CSF	144 mmc
Polymorphonuclear leukocytes in CSF	121 mmc
Proteins in CSF	915 mg/dL
Glucose in CSF	60 mg/dL
Albumin in CSF	480.6 mg/dL
Polymerase chain reaction (Biofire FILMARRAY) on CSF	Positive for *S. pneumoniae*

## Data Availability

The original contributions presented in this study are included in the article. Further inquiries can be directed to the corresponding author.

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
