# Peer review of "When Fever Strikes Twice: A Case Report of Streptococcus pneumoniae Myelitis with Delayed-Onset Reactive Arthritis"

_2036-7449, 2025, doi:10.3390/idr17060147_

Round 1
Reviewer 1 Report
Comments and Suggestions for Authors
This is an interesting case report of a complicated case of CNS Streptococcus pneumoniae infection, initially presenting as transverse myelitis caused by the detected pathogen and later complicated by reactive arthritis.
However, there is insufficient data to link the second manifestation to the patient's initial condition.
It is unclear from the patient's history whether other joint problems were present. Due to the patient's reduced mobility post the episode, this arthritis could easily have been triggered by stress.
CRP during the second epsode is not reported. No imaging is presented.
The figures need not include images (radiologic and not).
Author Response
Comment 1: "However, there is insufficient data to link the second manifestation to the patient's initial condition.
It is unclear from the patient's history whether other joint problems were present. Due to the patient's reduced mobility post the episode, this arthritis could easily have been triggered by stress."
Response 1: We thank the reviewer for this valuable comment.
In the revised version of the manuscript, we have expanded the discussion to provide a clearer explanation of the plausible immuno-mediated link between the initial pneumococcal myelitis and the subsequent reactive arthritis. Although a definitive causal relationship cannot be established, the temporal association, the sterile synovial fluid, and the exclusion of alternative infectious or autoimmune etiologies support a post-infectious immune mechanism.
Comment 2: "CRP during the second epsode is not reported. No imaging is presented."
Response 2: We have also added the C-reactive protein (CRP) value recorded during the arthritis episode, which showed a mild elevation, further supporting an inflammatory but non-septic process.
Comment 3: "The figures need not include images (radiologic and not)."
Response 3: Spinal MRI images have been included in the revised manuscript. The MRI, performed four days after the initiation of antibiotic therapy, showed no significant abnormalities—likely due to the timing of the scan and the early therapeutic response.
Reviewer 2 Report
Comments and Suggestions for Authors
This manuscript describes the first case of myelitis associated with arthritis following pneumococcal infection. For better publication quality, the authors could add relevant figures to the case, such as MRI or ultrasound images, to enrich the scientific article. Additionally, I suggest combining Tables 1 and 2.
Author Response
Comment 1: "For better publication quality, the authors could add relevant figures to the case, such as MRI or ultrasound images, to enrich the scientific article."
Response 1: We thank the reviewer for this constructive suggestion.
In the revised version, we have added spinal MRI images to visually support the clinical description. The MRI was performed four days after the initiation of antibiotic therapy and showed no significant abnormalities, likely due to the early treatment response.
Comment 2: "Additionally, I suggest combining Tables 1 and 2."
As recommended, we have also combined Tables 1 and 2 into a single comprehensive table to improve readability and coherence.
We appreciate the reviewer’s insightful comments, which have contributed to enhancing the quality and clarity of the manuscript.
Reviewer 3 Report
Comments and Suggestions for Authors
The colleagues from Italy report an interesting case of acute transverse myelitis (ATM) caused by Streptococcus pneumoniae, complicated by reactive arthritis. The case is both novel and educational. I have only a few major comments:
-
Please elaborate further on bacterial causes of ATM in your discussion and include Lyme neuroborreliosis presenting as ATM in the differential diagnosis. Support this section with relevant references on the topic. Lyme disease is particularly important to mention, as it can also cause arthritis; therefore, the differences between Lyme arthritis and reactive arthritis should be discussed.
-
The introduction would benefit from a brief description of life-threatening but fortunately rare complications of Streptococcus pneumoniae, such as purpura fulminans, which can occur even in immunocompetent individuals.
-
In the case presentation, please provide the normal reference values for CRP and procalcitonin at your institution, as these vary between countries and laboratories.
-
Do you have MRI images demonstrating the features of ATM to support your diagnosis? Including such imaging would strengthen the report.
-
The discussion should also address the use of intravenous steroids in cases of myelitis to reduce spinal cord swelling.
-
Finally, please indicate whether the patient experienced any long-term residual neurological deficits.
Author Response
Reviewer comment 1
“Please elaborate further on bacterial causes of ATM in your discussion and include Lyme neuroborreliosis presenting as ATM in the differential diagnosis. Support this section with relevant references on the topic. Lyme disease is particularly important to mention, as it can also cause arthritis; therefore, the differences between Lyme arthritis and reactive arthritis should be discussed.”
Response:
Thank you for this suggestion. We expanded the Discussion to explicitly enumerate bacterial causes of acute transverse myelitis (including Streptococcus pneumoniae, Neisseria meningitidis, Haemophilus spp., Staphylococcus aureus, Streptococcus agalactiae, Listeria monocytogenes, Borrelia burgdorferi, Brucella spp., and Mycobacterium tuberculosis) and summarized their mechanisms of spinal involvement (hematogenous seeding, contiguous spread, or immune-mediated injury). We added a focused paragraph on Lyme neuroborreliosis, noting that although myelopathy is rare, it is a recognized manifestation and may be associated with radiculitis/meningitis and arthritis. We contrasted Lyme arthritis (often oligo/monoarticular, potentially supported by serology or PCR/molecular detection in synovial fluid) with reactive arthritis (sterile post-infectious synovitis without direct pathogen detection), and we explained that in our patient the positive CSF PCR for S. pneumoniae, negative Lyme testing, and sterile synovial fluid favour a post-infectious reactive process rather than Lyme arthritis. Relevant references supporting these points were added.
Reviewer comment 2
“The introduction would benefit from a brief description of life-threatening but fortunately rare complications of Streptococcus pneumoniae, such as purpura fulminans, which can occur even in immunocompetent individuals.”
Response:
We added a succinct sentence in the Introduction noting that S. pneumoniae can cause life-threatening systemic complications (for example severe sepsis, disseminated intravascular coagulation and purpura fulminans), to emphasize the potential systemic severity of invasive pneumococcal disease.
Reviewer comment 3
“In the case presentation, please provide the normal reference values for CRP and procalcitonin at your institution, as these vary between countries and laboratories.”
Response:
We have included the institution’s reference ranges in the Case Presentation
Reviewer comment 4
“Do you have MRI images demonstrating the features of ATM to support your diagnosis? Including such imaging would strengthen the report.”
Response:
We included spinal MRI images (Figure 1) in the revised manuscript; MRI was performed after 4 days of treatment so it showed non-significant findings.
Reviewer comment 5
“The discussion should also address the use of intravenous steroids in cases of myelitis to reduce spinal cord swelling.”
Response:
We added a balanced paragraph on corticosteroid use in acute myelitis. The Discussion now notes that high-dose intravenous corticosteroids are commonly used in idiopathic and immune-mediated myelitides to reduce cord inflammation and oedema, but that in confirmed bacterial infections their use must be individualized and considered only in conjunction with adequate antimicrobial therapy due to potential immunosuppressive effects.
Reviewer comment 6
“Finally, please indicate whether the patient experienced any long-term residual neurological deficits.”
Response:
We have reported the longer-term outcome: at six-month follow-up the patient had regained full motor function with only mild residual urinary urgency and no new neurological relapses. T
Round 2
Reviewer 3 Report
Comments and Suggestions for Authors
Line 54-59 do not contain any reference to the statements made: suggest reviewing the following: Septic Shock and Purpura Fulminans Due to Streptococcus pneumoniae Bacteremia in an Unvaccinated Immunocompetent Adult: Case Report and Review - PMC and Purpura fulminans due to Streptococcus pneumoniae bacteraemia in an unsplectomised immunocompetent adult without primary hypocomplementaemia - PMC
Line 231 to 235 are lacking references on acute traverse myelitis due to Lyme neuroborreliosis
Author Response
Thank you for your valuable feedback.
Regarding lines 54–59, we acknowledge that the statements provided were not adequately referenced. We will revise this section and include the suggested articles, as well as any additional relevant literature, to strengthen the discussion.
Concerning the comment on lines 231–235, we would like to clarify that this section is part of the take-home messages, which are meant to summarise key concepts already discussed in the main text rather than introduce new content requiring additional references. Moreover, a reference specifically addressing Lyme neuroborreliosis was included earlier in the manuscript (line 168). For clarity, we will ensure that the flow between the main text and the take-home messages is more explicit, so that readers can easily trace back the supporting literature.
We appreciate your suggestions and will revise the manuscript to further improve its clarity and scientific rigor.